# A Parsonian Approach to Patient Safety: Transformational Leadership and Social Capital as Preconditions for Clinical Risk Management—the GI Factor

**DOI:** 10.3390/ijerph17113989

**Published:** 2020-06-04

**Authors:** Holger Pfaff, Jeffrey Braithwaite

**Affiliations:** 1Institute of Medical Sociology, Health Services Research and Rehabilitation Science (IMVR), Faculty of Human Science and Faculty of Medicine, University of Cologne, Eupener Str. 129, 50933 Cologne, Germany; 2Centre for Healthcare Resilience and Implementation Science, Australian Institute of Health Innovation, Macquarie University, 75 Talavera Road, Sydney, NSW 2109, Australia; jeffrey.braithwaite@mq.edu.au

**Keywords:** patient safety culture, clinical risk management, transformational leadership, social capital, Parsons, AGIL scheme, hospital, Germany

## Abstract

The purpose of this study is to investigate the role of the combination of transformational leadership and social capital in safety capacity building. Drawing on the A-G-I-L concept of Talcott Parsons, we test a model for patient safety. The hypothesis is, that good safety management needs a combination of goal attainment (G) and integration (I), here called the GI factor. We tested this hypothesis by using transformational leadership as a surrogate for goal attainment and social capital as a surrogate for integration in a study of the perceptions of chief medical officers in 551 German hospitals. We conducted a cross-sectional hospital survey combined with secondary data analysis in all German hospitals with at least one internal medicine unit and one surgery unit (*N* = 1224 hospitals) in the year 2008 with a response rate of 45% (*N* = 551). The regression model explained 17.9% of the variance in perceived clinical risk management. We found that if both requirements for goal-oriented collective action—transformational leadership and social capital—are met, good safety management is more likely. The tentative conclusion is that it takes transformative leaders and cohesive followers together as a social basis to improve safety in hospitals.

## 1. Introduction

Patient safety in hospitals has been researched extensively in recent years [1,2,3,4,5,6,7,8,9,10,11,12,13,14,15]. Nonetheless, we have comparatively few studies that try to uncover which organisational factors contribute to variation in hospital safety [1,2,3]. Where there is evidence, it mostly deals with safety-specific measures as features of quality management activities, e.g., accreditation and hand hygiene campaigns [6,7,8]. Compared to this there is a relative lack of research on the safety-unspecific factors. In this article we want to make a contribution to fill this research gap. To do this, we need non-safety related social theories to explain safety. 

In search for a mature theory that fits our problem, we went back to the structural-functional systems theory of Talcott Parsons. We especially stress the importance of his four-function paradigm, the AGIL scheme: adaptation (A), goal attainment (G), integration (I), and latent pattern maintenance (L). Applying this theory, we can immediately discern that effective quality and safety needs continuous adaptation over time to meet organisational and clinical demands (A), specific targets (G), across-the-board collaboration (I) and sufficient attention to underlying long-term system needs (L) [16,17,18]. In this paper we home in on two of these functions: G and I, because, combined, they are necessary to understand collective action. Based on this assumption, the goal of this paper is to test the hypothesis that good safety management needs a combination of goal attainment (proxy: transformational leadership) and integration (proxy: social capital); we call this the GI factor. 

## 2. The GI Factor: Collective Action by Transformational Leadership Combined with Cohesive Followers

To fulfil the function of *goal attainment* (the G-factor) with regard to the implementation of patient safety programs, there is a need for leadership that sets collective goals and inspires the followers to reach these goals. According to Bass and Riggio, transformational leadership is especially able to fulfil this condition [19]. Transformational leadership behaviours can strengthen the commitment of followers towards goals [19,20,21,22]. Literature shows that effective transformational leadership is associated with a rise in individual safety performance [23,24], front-line nursing managers’ perceptions of quality of care [25], clinical goal achievements [26], and quality improvement efforts in hospitals [27]. 

Deficits in integration (the I-factor) often lead to suboptimal patient care [28,29]. According to the social capital theory, achieving integration needs a certain amount of social capital [30,31,32,33,34,35,36]. Social capital facilitates “coordination and cooperation for mutual benefit” [31] (p. 69). We define social capital as communal social relationships characterized by common understanding, a good atmosphere, trust, we-feeling, mutual help, and reciprocity [37] as well as by commonly held values and beliefs [38]. Various studies show that social capital in healthcare organisations is associated with better coordination [39], work engagement [40], job satisfaction [41,42], health [43], and safety culture [44].

Our hypothesis is that the combination of goal attainment (G) and integration (I), here called the GI factor, is a prerequisite for high performance in healthcare organisations in general and in patient safety specifically [45]. This is due to the fact that effective collective action needs (a) transformative leaders as well as (b) cohesive followers. Transformational leaders have a vision and establish goals and high expectations. They convey believable information about goals to their followers and energize them to reach these goals. If these followers are characterized by high social capital, they are likely to act more uniformly, as a social unit. The GI factor combination produces collective energy within a group and gives this energy a direction, producing goal-oriented collective action. In other words, social capital bundles otherwise chaotic individual energies and transforms these into social energy. If this social energy is supported or nudged in the right direction by transformative leaders, then effective collective action, as is needed in patient safety management, can result. Thus, the GI factor could be regarded as goal-oriented social capital (see Table 1).

Based on this framework (see also [45]), we predict that hospitals, in which Chief Medical Officers (CMOs) report that their hospital is governed by transformational leaders leading cohesive followers, will perform well in “risk management”. We predict lower levels of perceived risk management in hospitals where the CMO reports a low level of transformational leadership and a low level of social capital. We also predict a medium level of risk management if only one of these two conditions applies. With this model, we will not only test our hypothesis regarding the GI factor but also a key element in Parsons’ theory. This is an important claim because critics have argued that Parsons’ theory is too abstract for empirical testing. We disagree.

## 3. Materials and Methods 

### 3.1. Design, Data Collection and Study Population

Data used here are survey data from the standardized cross-sectional study “Effects of hospital ownership structures on quality of healthcare” (HOSQua). The HOSQua research project was conducted by the Institute of Medical Sociology, Health Services Research and Rehabilitation Science, Faculty of Human Science and Faculty of Medicine, University of Cologne, Germany (IMVR). The research was conducted in collaboration with the Research Institute of the largest health insurance company in Germany, the Regional Healthcare Insurance (AOK and WidO) and funded by the German Medical Association. The survey and data collection process received approval from the Ethics Committee of the University of Cologne (ethical code: KöIn 22.02.2008).

In 2008, we conducted a cross-sectional study in all German hospitals with at least one internal medicine unit and one surgery unit (*N* = 1224 hospitals). The CMO of each study hospital was contacted and asked to respond to a postal survey with a standardized paper-and-pencil questionnaire. The survey was conducted from April to October 2008. We used the Total Design Method according to Dillman to maximise the response rate [46]. According to the 2008 German hospital quality report provided by the Federal Joint Committee (G-BA), 39.8% of these hospitals are public, 44.6% are not-for-profit hospitals (charitable hospitals), and 15.6% are for-profit hospitals. We used this 2008 German hospital quality report as an additional data source to measure structural characteristics of the hospitals (secondary data). 

We focused on CMOs as the key informants for medical quality in hospitals. CMOs are often used as participants in organisational research [47,48,49]. A benefit of enrolling CMOs is their “upper echelon” perspective [50]. Medical directors are knowledgeable and powerful actors both in care-related issues and in their feature as prominent leaders [51]. Some scholars have argued additionally that medical directors have referent power, i.e., regarding their capacity for “inspiration, motivation or identification” [52] of employees. We also regard medical directors as powerful key informants having the ability to implement structures and cultures that have the potential to improve quality and safety. Additionally, they are in positions where they have an appreciation for both the clinical and organizational aspects of patient safety.

### 3.2. Measures

All study measures were pre-tested [53] in order to avoid ambiguities with words or concepts. Pretesting included discussing the questionnaire with five experts from the fields of internal medicine, hospital quality management, and healthcare management. Afterwards, we conducted two think-aloud interviews with physician executives in order to fine-tune the final version of the questionnaire.

Transformational leadership: A six-item-scale was used to measure the variable transformational leadership (Cronbach’s α = 0.83) (see Table 2). This was a shortened form of the German version of the Transformational Leadership Inventory (TLI) [21,54]. Items in this scale represent one of the six key behaviours of transformational leaders identified by Podsakoff et al. (1990) [21]. The items with the highest factor loading on each scale of the German version [54] were used. The medical directors were asked to assess how often the executives of their hospital, e.g., top management, head physicians, and nursing management, exhibited transformational behaviour. A Likert scale was used ranging from “never” (1) to “always” (5). Examples of these items were: “The executive managers lead by example”, “The executive managers inspire others with their plans for the future”, and “The executive managers develop a team attitude and spirit among employees”. To be able to classify the hospitals as low and high transformational leadership hospitals (see Table 1), we dichotomized the scale using the median split method creating a binary variable: perceived low and high transformational leadership in the top management.

Social Capital: Based on the social capital theory [30,31,32,33,34,35,36,55,56,57] and the community concept [37,38,58], a communal social capital scale was constructed measuring six central aspects of this construct: common understanding, good working atmosphere, mutual trust, we-feeling, mutual help, and common values. This Social Capital of Organisations (SOCAPO) scale is a validated scale [59] and can be used to measure social capital of employees (SOCAPO-E) [59,60], social capital of hospital boards (SOCAPO-B) [61], and social capital of hospitals from the perspective of the CMO as shown in the present study (SOCAPO-CMO). Item examples are: “In our hospital, there is unity and agreement” and “In our hospital, we trust one another”. A Likert scale was used for the answers, ranging from “strongly disagree” (1) to “strongly agree” (4) [62]. Cronbach’s α for this scale is 0.88 (see Table 2). To be able to classify the hospitals as low or high social capital hospitals (see Table 1), we dichotomized the scale using median split, creating a binary variable: perceived low social capital in the organisation and perceived high social capital in the organisation.

Goal-integration factor: To test the combination of goal attainment and integration—the goal-integration factor—we created a “Goal-Integration Index” by summing the two categorical variables transformational leadership and social capital, deriving an index with three categories (0 = low transformational leadership plus low social capital; 1 = low transformational leadership plus high social capital or high transformational leadership plus low social capital; 2 = high transformational leadership plus high social capital). By doing this, we do not differentiate between the two combination of transformational leadership without social capital and social capital without transformational leadership. The reason is that both combinations are considered equivalent with regard to promoting collective action. For the present study, distinguishing between both combinations is deemed unnecessary.

Control Variables: We controlled for hospital size by using the quality reports of German hospitals. Hospital ownership was extracted from the 2008 German hospital quality report provided by the Federal Joint Committee (G-BA). Teaching status was measured by asking the CMO if their hospital was a “teaching hospital” or “non-teaching hospital”. We used teaching status as a proxy for up-to-date knowledge. We used hospital size (number of beds) as a continuous variable in our analyses (see Table 2). In the analysis, hospital ownership was recorded using the dummy variables charitable hospitals/not-for-profit ownership (1 = “Yes”; 0 = “No”) and private hospitals/for-profit ownership (1 = “Yes”; 0 = “No”). Public hospitals/public ownership was chosen as the reference category. This distinction was made to account for ownership effects.

Dependent variable “clinical risk management”: The core of clinical risk management is analysing the causes of errors and supporting efforts to limit the incidence of errors. Risk management has the broad goal of creating resilient systems with capacity to adapt, to learn and to tolerate the manifestation of errors [63]. Risk management is a key responsibility of all members in an organisation and thus a collective endeavour. In this study, clinical risk management was measured by a six-item scale (Cronbachs’ α: 0.78) (see Table 2). This scale was developed using cognitive pretests, exploratory factor analysis, and reliability tests [62] and applied in different studies (e.g., [60]). Item examples are: “Here in our hospital, everything is done to determine the causes of critical incidents that have occurred”, “Appropriate actions are always taken following a near-accident”, or “In our hospital, all nosocomial infections are reported”. All items were scored on a 4-point scale ranging from 1 (“strongly disagree”) to 4 (“strongly agree”). Inspection of residuals showed no considerable deviation from norms.

### 3.3. Analyses

First, we analysed the data using descriptive statistics, calculating frequencies and percentages for categorical variables as well as means and standard deviations for continuous variables. For multivariate analysis testing the GI hypothesis, we ran two regression models, a restricted model and a full model. In the restricted model, we entered the control variables reflecting the structural features of the hospitals. The full model additionally contained the central independent variable, the goal-integration index, as dummy variables. SPSS (version 26, IBM, Armonk, NY, USA) was used to conduct statistical analyses.

## 4. Results

### 4.1. Sample Characteristics

The response rate was 45.0% (551 out of 1224 questionnaires). Table 2 and Table 3 show the descriptive results for the sample and variables under study. A total of 551 included hospitals that had an average of 345 beds (SD: 282), ranging from 32 to 2322 beds indicating that some hospitals deviated considerably from the group mean. Among these hospitals, 41.5% (*N* = 228) were publicly funded, 46.9 % (*N* = 258) had a private not-for-profit ownership (charitable hospitals), and 11.6% (*N* = 64) were in private-for-profit ownership (private hospitals). Of the responding hospitals, 60.2% (*N* = 331) were non-teaching hospitals. The descriptive statistics of our continuous measures of transformational leadership (proxy for goal attainment) and social capital (proxy for integration) are displayed in Table 2. The results of the dichotomization of these variables are depicted in Table 3. On average, the medical directors rated the social capital of their hospitals as satisfactory to good (mean: 17.4 within a range from 8 to 24) with considerable variation (SD: 2.9). The dichotomization of this scale using median split allocated 42.9 % of the hospitals to low and 57.1% to high in social capital. CMO respondents rated transformational leadership in their hospital satisfactory (mean: 21.5 within a range from 7 to 30), again with considerable variation (SD: 3.3). The dichotomization of the transformational leadership scale via median split led to 45.6% of the hospitals being low and 54.4% being high on transformational leadership. The creation of the goal-integration index led to three GI levels, with 29.9% of the hospitals (*N* = 157) on the low level, 29.0% (*N* = 152) on the middle level, and 41.1% (*N* = 216) on the highest level (see Table 3).

### 4.2. Bivariate Analysis

The intercorrelations between the four continuous variables are presented in Table 4. The strongest correlations were between transformational leadership and social capital. The results show that perceived social capital as well as perceived transformational leadership are related to clinical risk management activities. We inspected multicollinearity using the condition index. None of the condition indices exceeded the value of 5, indicating that multicollinearity is not given. Applying t-tests, we found no significant difference between teaching hospitals and non-teaching hospitals regarding perceived clinical risk management, social capital, and transformational leadership. Teaching hospitals are significantly larger than non-teaching hospitals. With ANOVA, we found no significant differences between the three groups of ownership (public, private-not-for-profit (charitable), private-for-profit hospitals) in the continuous variables perceived social capital, transformational leadership, and clinical risk-management. There was no significant relationship between ownership and teaching hospital status. Public hospitals had significantly more beds than private-not-for-profit hospitals or private-for-profit hospitals. There was no relationship between ownership, and teaching status, on the one hand, and the GI factor, on the other hand. There was a significant negative relationship between number of beds (hospital size) and clinical risk management.

### 4.3. Regression Analysis

The results of the stepwise linear regression analysis are presented in Table 5. On the basis of the outlier statistics, we excluded one hospital from our sample due to large Cook’s distance. The restricted model explains 1.6% of the variance in perceived clinical risk management and was not significant. In the full model, we entered the goal-integration index (dummy variable with “goal-integration low” as reference category). The full model explains 17.9% of the variance in perceived clinical risk management. Hospitals that are high in goal-integration had significantly higher scores in perceived clinical risk management (2.42 units higher on a scale from 6 to 24) compared to the reference group, which consists of the hospitals with the lowest goal-integration level. Similarly, hospitals with a medium goal-integration level had a significant higher score (1.1 units higher) on the perceived clinical risk management scale than hospitals with the lowest goal-integration level, although—as predicted—the difference is not as large as with hospitals scoring high on the GI index. The residuals statistics (e.g., Q-Q-plot of the standardized residuals) was unremarkable.

## 5. Discussion

### 5.1. The Study Aims in Context

Deficits in quality and patient safety in complex hospitals present a major risk to hospital patients and may have adverse outcomes. We used a theory capable of providing a framework to make this situation theoretically accessible, an approach stressing the importance of necessary preconditions for organisational performance. The structural functional approach of Talcott Parsons is an appropriate theory for this, especially his AGIL scheme. This theory enables us to connect social theory with quality and safety research and to integrate existing knowledge about transformational leadership and social capital. Until now, little was known about how transformational behaviour can foster quality and safety and especially risk management in hospitals in combination with other variables. There was a deficit in applying the structural functional theory to patient safety research in the past. The same applies to social capital and transformational leadership. To the best of our knowledge, applying the combination of these concepts is a novel feature in quality and safety research.

The empirical results suggest a strong, significant relationship between the goal-integration dimension and how people deal with patient safety, namely through managing clinical risks. The perceived transformational behaviour of the top management in hospitals combined with a high level of perceived social capital of the followers within the hospital is connected with higher levels of perceived safety activities compared to hospitals with low levels of transformational leadership and low levels of social capital. This finding provides useful insights into a type of leader–follower interaction, which is conducive to quality and safety in hospitals. The quintessence is: “it takes two to tango”, i.e., it takes transformative leaders and cohesive followers. If the followers are not integrated and cohesive, they are not able to act as a sufficiently aggregated unit and thus—to extend the metaphor—are not able “to dance” coherently with the leader. Instead, they are disconnected, do not appreciate the goal, give up in confusion, or simply struggle. Dancing the safety-tango takes a committed two: the goal-attaining leader and the integration-oriented followers.

It follows from this that it is charisma and inspiration rather than sanctions and punishment that leaders should use to motivate their followers [27,64]. Rather than social conflicts about values and competition, it is due to social cohesion and social solidarity that teams, departments, and hospitals need to facilitate collective activity like those inherent in quality management. If both come together, synergistic effects can be encouraged to emerge in a marriage of visionary leaders and cohesive followers.

In its broader sense, we call this the goal-integration-factor in safety or, in short, the GI factor. We are persuaded that the AGIL scheme of Parsons delivers important indications regarding fruitful system preconditions for safety. According to these results, the “GI” in the AGIL scheme represents an extremely important component of safe care. 

The findings not only correspond to the structural functional theory of Talcott Parsons but also to concepts of social capital, too. There are at least three pathways which explain how social capital is built and connected to safety and transformative leadership. The first is the values–social capital pathway. This pathway starts with common values, which are the basis for trust. Trust itself is necessary to start and maintain social reciprocity. This leads to mutual obligations, transforming groups into cohesive units high in social capital. Such socially integrated groups are capable of acting effectively as a collectivity [55,65]. The second pathway is the social capital–performance–social capital pathway [66]. It starts with the social capital of the group, which facilitates collective action by producing conformity. This fosters the capability to act coherently as a group, resulting in an accumulation of resources by the group. In turn, this performance makes the group even more attractive for its members. Consequently, the social capital of the group increases as a result of this positive feedback loop. This can lead to an upward spiral. In a negative scenario, a downward spiral would emerge, leading to a diminishment of organisational and personal resources in the long run [67]. The third pathway is the social capital–safety pathway. In its basic form, this pathway comprises the hypothesis that cohesion and trust, the core of social capital, enable open communication about safety issues. This openness in communication is a general precondition for safety culture. Within such a culture, employees can speak about their mistakes more openly without being sanctioned by the leader or the group. 

But it is not just collaboration that is key. Leadership is pivotal and is always in the frame. Transformative leaders can use all three pathways to facilitate collective action capacity and to use this bundled social energy to meet organizational goals. The collective force of teams and staff can be harnessed by a transformative leader in order to build forward momentum in creating safer care and enhanced quality. It takes someone in a committed leadership position to grasp the opportunities these insights afford, and to transform the often latent collective energy. We must bear in mind, as many others have reported, that barriers to front line participation in quality activities and, for example, to reporting incidents and adverse events, include perceived or actual sanctions or fearfulness or a combination of these. Effective transformational leadership can support employees in overcoming group- or self-interests for the benefit of safer, higher quality care. A mature culture of reporting, for example, needs effective leader–follower relations and widespread organisational trust [21,24,68]. 

### 5.2. Limitations

The study results are subject to limitations. Firstly, the analyses are based on cross-sectional data, which did not allow causal conclusions. We would support longitudinal studies on this topic. A second limitation is the response rate. Although this is a more than acceptable response rate for surveys of physicians or senior managers [48,69,70], we do not know about non-respondents. Studies of this kind cannot eliminate the possibility of bias due to non-response. Thirdly, the dependent and the independent variables stem from the same data source, assessed with the same method. This may cause common method variance. Mitigating this, there was no evidence of multicollinearity, and additionally, not all control variables were gathered by the survey of the CMOs (e.g., hospital size and ownership status) and are thus not subject to this form of bias. Fourthly, the source of information, the medical director, was a key informant [71] representing the hospital. The data mirrors the medical directors’ perception of the top managers’ leadership style, the hospitals’ social capital, and clinical risk management. This should be kept in mind. A fifth limitation lies in the measurement of transformational leadership. The scale used for measuring transformational behaviour consisted of one item for each of the underlying behavioural dimensions [21]. The original studies assessed transformational behaviour using 26 items in the German version [21,54]. For practical reasons, we used a short form of the transformational leadership scale with satisfactory reliability. Further research should validate the construct and criterion validity of this scale more extensively. The sixth limitation lies in the measurement of social capital. The scale used for measuring social capital consisted of one item for each of the underlying dimensions of community based social capital [37,55,59,72]. The reliability of the social capital scale is good. A seventh limitation lies in the measurement of clinical risk management. The scale consists of six items measuring different aspects of clinical risk management with satisfactory levels of reliability. Subsequent research should further validate the construct and criterion validity of this short scale. The use of a self-reported measure of clinical risk management could be affected by biases such as social desirability responses. The eighth limitation is connected with the year of data gathering (2008). This does not necessarily limit its usefulness, because we measured general, time-independent resistance resources of healthcare organizations (G + I). The theory of Talcott Parsons is a time-independent theory. Additionally, we think that these resources are quite stable over time and, more importantly, that their impact on safety culture does not erode or vanish as time goes by. The magnitude of the impact could be subject to alterations, but the effect itself will remain. We strongly believe therefore that the understanding and perception of the key concepts of Parsons, the way they were measured, and their possible impact on risk management remain relevant to today’s context.

## 6. Conclusions

The results suggest that self-reported goal attainment and social integration together correlate with collective behaviours in favour of safer care, quality initiatives, and the management of clinical risk. This was specified in the present study by measuring self-reported transformational leadership combined with self-reported social capital and perceived risk management. This study provides insights into the relationship between this combination of these factors as reported by a large cohort of medical directors. Medical directors who rate the transformational leadership and social capital of their hospitals highly are more likely to perceive better management of clinical risks, a core element in safe care, and better quality of care. This result was partly a test of the AGIL scheme of Talcott Parsons’ structural functional systems theory.

This research may have practical implications. First, hospital decision makers might consider investing in transformational leadership behaviours of their leaders by creating a combination of transformational leadership training and peer consulting. Secondly, hospital decision makers should consider enhancing their hospital’s social capital to influence quality and safety performance among members of staff, different professionals, and different departments. This can be done, for example, by establishing official and informal rules and norms; promoting trust by creating transparency, reciprocity, and mutual help; and by emphasizing the “we” and not the “I” in work, thereby reducing unnecessary competition between employees. To further improve the hospital’s social capital, managers could follow the three suggestions of Cohen and Prusak (2001) [72]: making connections, enabling trust, and fostering cooperation. Making connections refers to retaining employees, enabling networking, and giving employees the opportunity and the time to interact socially to foster bonding. Enabling trust starts with a management that trusts its employees and that attaches less importance to controlling things and people. To foster cooperation, it is most helpful to give employees “a common sense of purpose” [72]. The top executives should additionally emphasize a sense of unity, we-feeling and affiliation.

## Figures and Tables

**Table 1 ijerph-17-03989-t001:** The goal-integration model: The GI factor as goal-oriented social capital.

	Integration = Low	Integration = High
**Goal attainment = High**	Goal-oriented individual capital	Goal-oriented social capital (GI factor)
**Goal attainment = Low**	Non-goal-oriented individual capital	Non-goal-oriented social capital

**Table 2 ijerph-17-03989-t002:** Statistical description of independent and dependent variables (*N* = 551).

Variable	*N*	Items	Range	Mean	SD	Alpha
Perceived social capital of hospital (SC)	536	6	8–24	17.4	2.9	0.88
Perceived transformational leadership (TL)	539	6	7–30	21.5	3.3	0.83
Perceived clinical risk management (CRM)	539	6	7–24	18.7	2.6	0.79
Number of beds	550	-	32–2322	344.6	282.0	-

**Table 3 ijerph-17-03989-t003:** Characteristics of hospitals in the analysis (*N* = 551).

Construct	Categories	*N*	%
Ownership (missing: *N* = 1)	Public hospitals	228	41.5
	Private-not-for-profit hospitals (charitable hospitals)	258	46.9
	Private-for-profit-hospitals	64	11.6
Teaching status (missing: *N* = 1)	Teaching hospital	219	39.8
	Non-teaching hospital	331	60.2
Perceived social capital of hospital (SC) (missing: *N* = 15)	Low	230	42.9
	High	306	57.1
Perceived transformational leadership in hospital (TL) (missing: *N* = 12)	Low	246	45.6
	High	293	54.4
Goal-integration (missing. *N* = 26)	Goal-integration low (social capital low + transformational leadership low)	157	29.9
	Goal-integration middle (social capital low + transformational leadership high OR social capital high + transformational leadership low)	152	29.0
	Goal-integration high (social capital high + transformational leadership high)	216	41.1

**Table 4 ijerph-17-03989-t004:** Pearson’s correlations between the continuous variables under study (*N* = 551; missing: 36).

	Perceived Social Capital of Hospital	Perceived Transformational Leadership	Number of Beds	Perceived Clinical Risk Management
Perceived social capital of hospital	1	0.601 **	−0.125 **	0.431 **
Perceived transformational leadership	0.601 **	1	−0.036	0.438 **
Number of beds	−0.125 **	−0.036	1	−0.13 **
Perceived clinical risk management	0.431 **	0.438 **	−0.133 **	1

** The correlations are significant at the p-level of 0.01 (two-sided). listwise deletion *N* = 515; missing: 36.

**Table 5 ijerph-17-03989-t005:** Regression analysis with perceived clinical risk management as dependent variable (restricted and full model, *N* = 514).

	Restricted Model	Full Model
Constant	18.811	<0.000	(18.33, 19.30)	17.469	<0.000	(16.92, 18.02)
Private-for-profit hospitals(reference category: public hospitals)	0.620	0.097	(−0.11, 1.35)	0.546	0.110	(−0.13, 1.22)
Private-not-for-profit hospitals (charitable hospitals) (reference category: public hospitals)	0.246	0.305	(−0.23, 0.72)	0.185	0.401	(−0.25, 0.62)
No. of beds	−0.001	0.077	(−0.002, 0.000)	−0.001	0.220	(−0.001, 0.000)
Teaching hospital (yes: 1; no: 0)	0.039	0.877	(−0.45, 0.53)	−0.040	0.861	(−0.49, 0.41)
GI = 1 (transformational leadership = high and social capital = low or vice versa) (Reference category: GI = 0)				1.095	<0.000	(0.58, 1.61)
GI = 2 (transformational leadership = high and social capital = high)(Reference category: GI = 0)				2.418	<0.000	(1.94, 2.90)
Explained Variance	0.016	0.083		0.179	<0.000	
Change in explained variance: restricted model —full model				0.163	<0.000

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
