# Peer review of "A Parsonian Approach to Patient Safety: Transformational Leadership and Social Capital as Preconditions for Clinical Risk Management—the GI Factor"

_ijerph, 2020, doi:10.3390/ijerph17113989_

Round 1

Reviewer 1 Report

The study is interesting. The topic the authors have selected is relevant.

It is necessary to precise, in the introduction, the aim of the study and the purpose the authors propose to achieve investigating the subject (even if it is done in the abstract).

There is a lot of information included in the introduction, but it may be better placed in other section ("theoretical section”). Too much information is included in the introduction.

The research is appropriately designed and well presented.

One of my major concerns is related to the data that are old – from 2008 – and we are in 2020, for this reason results might be not so relevant. I believe that the responses received more than 10 years later could be different, mainly these related to social capital. Moreover, its  measurement is not convincing. That is the reason why I chose the option: "Reconsider after major revision" (lack of actuality), which is the most regrettable because the idea of this study is thoughtful and intriguing.

Author Response

Thank you very much for your important and good comments. Here are our responses.

Point 1. It is necessary to precise, in the introduction, the aim of the study and the purpose the authors propose to achieve investigating the subject (even if it is done in the abstract).

Response to point 1: We have rearranged the introduction getting faster to the purpose of the study, e.g.:

“1. Introduction

Patient safety in hospitals has been researched extensively in recent years [1-15]. Nonetheless we have comparatively few studies that try to uncover which organisational factors contribute to variation in hospital safety [1,2,3]. Where there is evidence, it mostly deals with safety-specific measures as features of quality management activities, e.g. accreditation and hand hygiene campaigns [6–8]. Compared to this there is a relative lack of research on the safety-unspecific factors. In this article we want to make a contribution to fill this research gap. To do this we need non-safety related social theories to explain safety.

In search for a mature theory which fits our problem, we went back to the structural-functional systems theory of Talcott Parsons. We especially stress the importance of his four-function paradigm, the AGIL scheme: adaptation (A), goal attainment (G), integration (I), and latent pattern maintenance (L). Applying this theory, we can immediately discern that effective quality and safety needs continuous adaptation over time to meet organisational and clinical demands (A), specific targets (G), across-the-board collaboration (I) and sufficient attention to underlying long-term system needs (L) [16-18]. In this paper we home in on two of these functions: G and I, because, combined, they are necessary to understand collective action. Based on this assumption the goal of this paper is to test the hypothesis that good safety management needs a combination of goal attainment (proxy: transformational leadership) and integration (proxy: social capital); we call this the GI factor.”

Point 2. There is a lot of information included in the introduction, but it may be better placed in other section ("theoretical section”). Too much information is included in the introduction.

Response to point 2: We have shortened the introduction and the explanation of theory substantially by canceling 63 lines (former: line 31-148; now: line 31-85) with now 1,25 pages (introduction plus theory).

Point 3. One of my major concerns is related to the data that are old – from 2008 – and we are in 2020, for this reason results might be not so relevant. I believe that the responses received more than 10 years later could be different, mainly these related to social capital. Moreover, its  measurement is not convincing. That is the reason why I chose the option: "Reconsider after major revision" (lack of actuality), which is the most regrettable because the idea of this study is thoughtful and intriguing.

Response to point 3: This is a good and important point. Our response is:

First, these data form 2008 have been the only data where we got all things together to test the hypothesis of Talcott Parsons about the relevance of goal attainment and social integration across the German hospitals. This was a singular chance.

Secondly, we think that the theory of Talcott Parsons is time-independent, focusing on general resistance resources of healthcare organisations. These AGIL-features could be subject to minor changes in magnitude and percentage over time, but the interrelationship between them and the outcomes are quite stable, in our opinion. This is a hidden general assumption in healthcare research because scholars cite research results older than 12 years quite often, thus implicitly admitting that these results are still relevant today. Therefore, if the AGIL resources are time-independent and if their impact on healthcare organisations outcomes is also quite time-independent, we would guess that survey data from a recent year make the results not really different. 

But because of the importance of your critique we added limitation 8 into the limitation section: 

The eighth limitation is connected with the year of data gathering (2008). This does not necessarily limit its usefulness, because we measured general, time-independent resistance resources of healthcare organizations (G + I). The theory of Talcott Parsons is a time-independent theory. Additionally, we think that these resources are quite stable over time and – more importantly – that their impact on safety culture doesn´t erode or vanish as time goes by. The magnitude of the impact could be subject to alterations, but the effect itself will remain. We strongly believe therefore that the understanding and perception of the key concepts of Parsons, the way they were measured and their possible impact on risk management remain relevant to today's context.

Reviewer 2 Report

Dear authors,

This is quite a long article that I took time to go through it repeatly. Based on the structure-functional system theory of Talcott Parsons, the authors tested the hypothesis that "good safety management needs a combination of goal attainment (proxy: transformational leadership) and integration (proxy: social capital), or the GI factors. The study concluded that if transformational leadership and socal capital are met, good safety management is more likely.

My comments are:

  1. This manuscript is well done, adopting social science theory into patient safety research field. To my knowledge, this is innovative. The authors explained the theory background in Introduction section. Given the audience of this Journal may not all have strong social science background, I would suggest authors to make it concise so that the audience can go to the Material and Methods without taking too much time in the Background.
  2. In terms of the data source, the authors used the survey data of 2008, which had only 45% response rate. Can authors explain why use this old data? Will survey data from another recent year make the results different?

Author Response

Thank you very much for your important and valuable comments. Here are our responses.

Point 1: This manuscript is well done, adopting social science theory into patient safety research field. To my knowledge, this is innovative. The authors explained the theory background in Introduction section. Given the audience of this Journal may not all have strong social science background, I would suggest authors to make it concise so that the audience can go to the Material and Methods without taking too much time in the Background.

Response to point 1: We shortened the introduction substantially by canceling 63 lines (former: line 31-148; now: line 31-85) with now 1,25 pages for introduction and theory.

Point 2: In terms of the data source, the authors used the survey data of 2008, which had only 45% response rate. Can authors explain why use this old data? Will survey data from another recent year make the results different?

Response to point 2: This is a good and critical point, indeed.

First, these data form 2008 have been the only data were we got all things together to test the hypothesis of Talcott Parsons about the relevance of goal attainment and social integration across the German hospitals. This was a singular chance.

Secondly, our answer to the important question of reviewer 2 – Will survey data from another recent year make the results different? - is: probably not really. Why? We think that the theory of Talcott Parsons is time-independent, focusing on general resistance resources of healthcare organisations. These AGIL-features could be subject to minor changes in magnitude and percentage over time, but the interrelationship between them and the outcomes are quite stable, in our opinion. This is a hidden general assumption in healthcare research because scholars cite research results older than 12 years quite often, thus implicitly admitting that these results are still relevant today. Therefore, if the AGIL resources are time-independent and if their impact on healthcare organisations outcomes is also quite time-independent, we would guess that survey data from another recent year make the results not really different. 

Because of the importance of reviewers 2 critique we added limitation 8 into the limitation section:  

“The eighth limitation is connected with the year of data gathering (2008). This does not necessarily limit its usefulness, because we measured general, time-independent resistance resources of healthcare organizations (G + I). The theory of Talcott Parsons is a time-independent theory. Additionally, we think that these resources are quite stable over time and – more importantly – that their impact on safety culture doesn´t erode or vanish as time goes by. The magnitude of the impact could be subject to alterations, but the effect itself will remain. We strongly believe therefore that the understanding and perception of the key concepts of Parsons, the way they were measured and their possible impact on risk management remain relevant to today's context.

Thirdly, we think that the response rate in this survey is in the range of other physician surveys and a bit below the general mean of 54% of physician surveys (Kellerman & Herold 2001) and substantially above the mean of 32% normally given in executive surveys (Cycyota & Harrison 2006; Bendar & Westphal 2006).

Literature:

Bednar, M. K., & Westphal, J. D. (2006). Surveying the corporate elite: Theoretical and practical guidance on improving response rates and response quality in top management survey questionnaires. Research methodology in strategy and management, 3, 37-56.

Cycyota, C. S., & Harrison, D. A. (2006). What (not) to expect when surveying executives: A meta-analysis of top manager response rates and techniques over time. Organizational Research Methods, 9(2), 133-160.

Kellerman, S. E., & Herold, J. (2001). Physician response to surveys: a review of the literature. American journal of preventive medicine, 20(1), 61-67.

Reviewer 3 Report

Thank you for the opportunity to review such an interesting and thought-provoking article. The study is well conceived, well written and highly original. I have only a couple of very minor comments and suggestions for the authors.

  • The survey data is 12 years old. Although this does not necessarily limit its usefulness, I felt that a stronger argument could have been made that the understanding and perceptions of the key concepts and how they were measured remain relevant to today's context.
  • Line 120 - should safe be safety?
  • The introduction argues that transformational leadership and social capital (GI factor) facilitate a good safety climate. However, the study can only demonstrate that perceptions of these functions/characteristics tend to occur together. The data cannot and does not show a causal relationship. This is both a limitation of the study and an (exciting) area for further research. The first sentence of the conclusion implies that the results demonstrate a causal link, i.e. that one influences the other, and should perhaps be re-worded.

Author Response

Thank you very much for your important, valuable and positive comments. Here are our responses.

Point 1: The survey data is 12 years old. Although this does not necessarily limit its usefulness, I felt that a stronger argument could have been made that the understanding and perceptions of the key concepts and how they were measured remain relevant to today's context.

Response to point 1: This is a good point. We therefore added limitation 8 to the limitation section including these sentences:

“The eighth limitation is connected with the year of data gathering (2008). This does not necessarily limit its usefulness, because we measured general, time-independent resistance resources of healthcare organizations (G + I). The theory of Talcott Parsons is a time-independent theory. Additionally, we think that these resources are quite stable over time and – more importantly – that their impact on safety culture doesn´t erode or vanish as time goes by. The magnitude of the impact could be subject to alterations, but the effect itself will remain. We strongly believe therefore that the understanding and perception of the key concepts of Parsons, the way they were measured and their possible impact on risk management remain relevant to today's context.

Point 2: Line 120 - should safe be safety?

Response to point 2: This hint is good. We skipped this line because two reviewers urged us to shorten the introduction section and reduced it to “and safety culture (44)” (line 65 in the new version)

Point 3: The introduction argues that transformational leadership and social capital (GI factor) facilitate a good safety climate. However, the study can only demonstrate that perceptions of these functions/characteristics tend to occur together. The data cannot and does not show a causal relationship. This is both a limitation of the study and an (exciting) area for further research. The first sentence of the conclusion implies that the results demonstrate a causal link, i.e. that one influences the other, and should perhaps be re-worded.

Response to point 3: That is a good and important point. We changed the first sentence in the conclusion in (line 347-350 in the new manuscript):

“The results suggest that self-reported goal attainment and social integration together correlate with collective behaviours in favour of safer care, quality initiatives and the management of clinical risk. This was specified in the present study by measuring self-reported transformational leadership combined with self-reported social capital and perceived risk management.”

Additionally, we altered the second sentence in the limitation section as follows:

“Firstly, the analyses are based on cross-sectional data, which didn´t allow causal conclusions.”

Round 2

Reviewer 1 Report

Thank you for the Authors' responses.

I did not mean to make the explanation of theory shorter, I only suggested to move the theoretical part from the introduction to another (autonomic) section. However, if the Authors want to reduce this part, it is their decision.

Because of adding the section 2 (“The GI factor: collective action by transformational leadership combined with cohesive followers”), “Materials and Methods” should be the section 3. Other sections and subsections ought to be renumerated.

I find the addition of the eighth limitation appropriate. It is a good solution concerning the old data.

Author Response

Thank you very much for this quick reply! Here are our answers:

Point 1: “I did not mean to make the explanation of theory shorter, I only suggested to move the theoretical part from the introduction to another (autonomic) section. However, if the Authors want to reduce this part, it is their decision.”

Response to point 1: Sorry for this misunderstanding. Because one of the other reviewers recommended to shorten introduction and theory (too), we thought that the length of the first part was a common problem. Therefore, we decided to abbreviate the introductory text. Thank you for your understanding.

Point 2: “Because of adding the section 2 (“The GI factor: collective action by transformational leadership combined with cohesive followers”), “Materials and Methods” should be the section 3. Other sections and subsections ought to be renumerated.”

Response to point 2: Thank you for pointing to this error. We renumerated the sections.

Point 3: “I find the addition of the eighth limitation appropriate. It is a good solution concerning the old data.”

Response to point 3: Thank you for accepting this solution concerning the old data.